# Highly efficient broadband terahertz generation from ultrashort laser filamentation in liquids

Indranuj Dey [1], Kamalesh Jana[1], Vladimir Yu. Fedorov[2,3], Anastasios D. Koulouklidis [4], Angana Mondal[1], Moniruzzaman Shaikh[1], Deep Sarkar[1], Amit D. Lad[1], Stelios Tzortzakis [2,4,5], Arnaud Couairon[6] & G. Ravindra Kumar[1]

Generation and application of energetic, broadband terahertz pulses (bandwidth ~0.1–50 THz) is an active and contemporary area of research. The main thrust is toward the development of efficient sources with minimum complexities—a true table-top setup. In this work, we demonstrate the generation of terahertz radiation via ultrashort pulse induced filamentation in liquids—a counterintuitive observation due to their large absorption coefficient in the terahertz regime. The generated terahertz energy is more than an order of magnitude higher than that obtained from the two-color filamentation of air (the most standard table-top technique). Such high terahertz energies would generate electric fields of the order of MV cm$^{-1}$, which opens the doors for various nonlinear terahertz spectroscopic applications. The counterintuitive phenomenon has been explained via the solution of nonlinear pulse propagation equation in the liquid medium.

[1] Tata Institute of Fundamental Research, 1 Dr. Homi Bhabha Road, Mumbai, MH 400005, India. [2] Science Program, Texas A&M University at Qatar, P. O. Box 23874 , Doha, Qatar. [3] P. N. Lebedev Physical Institute of the Russian Academy of Sciences, 53 Leninskiy Prospekt, 119991 Moscow, Russia. [4] Institute of Electronic Structure and Laser, Foundation for Research and Technology Hellas, P.O. Box 1527 , 71110 Heraklion, Greece. [5] Department of Material Science and Technology, University of Crete, P.O. Box 2208 , 71003 Heraklion, Greece. [6] Centre de Physique Théorique, École Polytechnique, CNRS, F-91128 Palaiseau, France. Correspondence and requests for materials should be addressed to G.R.K. (email: grk@tifr.res.in)

Energetic, broadband terahertz pulses generated by compact, table-top femtosecond (fs) lasers can drive major advances in ultrafast dynamics, nonlinear terahertz optics, and bio-material imaging[1-4]. Over the years, several approaches were undertaken to develop compact, coherent, tuneable terahertz radiation sources using interaction of short-pulse laser with various targets[5-8]. Among these, terahertz generation via two-color filamentation in gases is the most popular mechanism due to the ability of scaling the terahertz energy and wavelength through control of the input laser parameters[9-12]. However, this scaling is limited by the laser pulse energy, motivating the search for alternative mechanisms, either requiring crystals[13, 14], or vacuum technologies[15, 16]. Exploration of other simpler means of high energy, broadband terahertz generation is an active area of research.

Intense laser interaction with liquids has been studied by various researchers interested in filamentation in condensed media[17]. A plethora of nonlinear phenomena observed during filamentation in gases, like self-phase modulation, wave-mixing, supercontinuum generation, pulse compression etc., become much more pronounced in liquids due to the orders of magnitude higher nonlinear susceptibility and larger neutral density ($>10^{21}$ cm$^{-3}$) compared to gases[18, 19]. Therefore, one may expect that physical mechanisms like symmetry-broken current generation[9, 20], multi-wave mixing[21, 22], Raman downshifting[23], and Cherenkov emission[24] would be magnified in liquids leading to strong terahertz generation. Surprisingly, there has been little effort to explore this potential, owing primarily to the fact that liquids present strong absorption in the terahertz regime[25, 26]. However, if the generation mechanism is efficient enough, residual terahertz radiation may emerge from the liquid medium despite the absorption.

Here, we unravel the full potential of broadband terahertz generation in liquids under intense, femtosecond laser excitation. We study the dependence of the terahertz yield on input laser intensity, determine the polarization of the generated terahertz radiation and measure its spectrum. We show that using single-color filamentation in liquids, one can generate ultra-broadband terahertz radiation (bandwidth ~100 THz) with energies up to 76 μJ/pulse (at laser energy ~28 mJ), that are an order of magnitude higher compared to two-color filamentation in gases. Our simulations show that due to extremely efficient spectral broadening in liquids, significant laser energy transfers to frequencies around the pulse second harmonic, which in turn leads to the formation of an asymmetric plasma current and thereby generation of terahertz radiation in a way similar to the two-color filamentation scheme in gases. Finally, we demonstrate the potential of our source for nonlinear terahertz optics studies, by conducting an experiment on intensity-dependent nonlinear transmission of the generated terahertz radiation in a doped silicon wafer. It is observed that under the same laser input and terahertz focus conditions, the terahertz radiation generated from the liquid by single-color filamentation induces two times larger terahertz transmission than that due to terahertz produced from air by standard two-color filamentation.

## Results

### Setup for terahertz generation and detection.
The schematic of the experimental setup is shown in Fig. 1. An 800 nm, 48 fs laser pulse (peak energy ~5–50 mJ, 10 Hz repetition rate) is focused by a lens with focal length $f = 500$ mm into a 50 mm long infrasil cuvette, filled with the target liquid. The radiation emitted during filamentation inside the liquid is collected and guided onto a broadband pyroelectric detector (PED) (Gentec-THZ-5I-BNC-BL) located at the point F$_1$. Various filters are used to cut off the

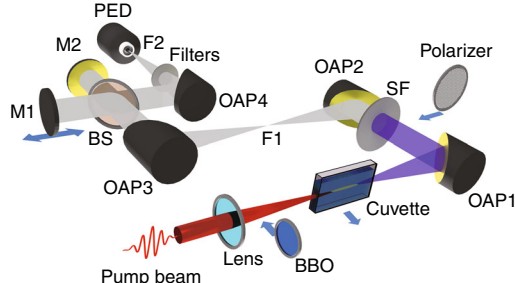

**Fig. 1** Experimental layout for integrated and spectral terahertz measurements. Broadband terahertz radiation is generated from the liquid in the cuvette via laser-induced filamentation. The terahertz radiation is collected and collimated by an off-axis parabola (OAP$_1$) filtered by a silicon filter (SF). The terahertz beam is then focused onto a PED by OAP$_2$ at focus F$_1$ for integrated measurements. A combination of filters is used before PED to minimize mid-IR radiation. A wire-grid polarizer can be inserted between OAP$_1$ and OAP$_2$, for terahertz polarization measurements. For spectrum measurements, the PED is removed from F$_1$, and the terahertz beam is steered using OAP$_3$ into a Michelson interferometer (consisting of mirrors M$_1$, M$_2$, and pellicle beam-splitter BS) for FAC measurements using the same PED placed at F$_2$ (focus of OAP$_4$)

visible and mid-infrared (mid-IR) radiation (see Methods). To compare our signal to the two-color filamentation in air, we remove the cuvette and place a 0.2 mm thick beta-barium borate (BBO) (type-I) crystal in the beam path (~400 mm from the lens).

### Energy and polarization measurements.
The integrated broadband terahertz energy measured for various liquids as a function of laser energy is shown in Fig. 2a. The reported energy is corrected for the terahertz transmittance of the filters and spectral power content in 0.1–50 THz range (see Methods). For comparison, on the same figure we plot the terahertz energy from two-color filamentation in air. An integrated broadband terahertz energy of ~76 μJ was obtained for laser energy of 28 mJ, which is about 20 times larger than what is obtained at similar energies during two-color filamentation in air, demonstrating a conversion efficiency $>10^{-3}$. For acetone, terahertz signals above the noise level of the PED with the mid-IR rejecting filter combination is obtained for an input laser pulse energy of as low as 0.1 mJ.

Figure 2b shows the integrated broadband terahertz far-field polarization plots for acetone and air for input laser energy $U_L$ ~10 mJ. The plot is obtained by inserting a wire-grid polarizer (Specac GS57204) in the collimated terahertz beam path between the off-axis parabolas OAP$_1$ and OAP$_2$ (Fig. 1), and recording the integrated terahertz energy as a function of the polarizer rotation angle. The plot for the two-color air terahertz shows a typical dipole pattern observed previously[27]. The broadband terahertz radiation from liquids also exhibits similar dipole-like characteristics, with the axis of polarization almost perpendicular to the input polarization. The contrast ratio between the dipole maxima to minima is ~1.2 for both air and acetone, indicating strong depolarization.

To understand the polarization of the resultant broadband terahertz emission, a study of the post-filamentation polarization states of the fundamental (800 nm), and the second harmonic (400 nm) part of the generated supercontinuum in air and liquid (acetone) is carried out (see Methods). It is observed that the 800 nm pulse suffers strong depolarization after filamentation in both air and acetone. Similar observation has been reported by Yu et al.[28], and has been attributed to the magnification of polarization perturbations (induced by optical elements) during nonlinear propagation process. The 400 nm polarization is found

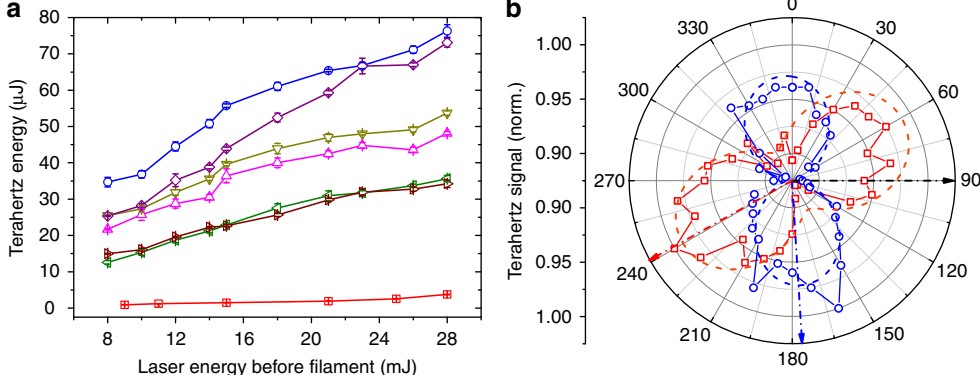

**Fig. 2** Broadband terahertz energy and polarization measurements by PED. **a** Energy measured by the PED (with filter and spectral corrections) from two-color filamentation in air (red square) and single color filamentation in various liquids: ethanol (magenta up-triangle), methanol (olive down-triangle), acetone (blue circle), dichloroethane (purple diamond), deionized water (dark-red right-triangle), carbon disulfide (green left-triangle), as a function of laser energy. Each data point is averaged over 32 acquisitions and the error bars indicate the standard deviation from the average. For an input laser energy of 28 mJ before the cuvette, an integrated terahertz energy of 76 μJ is obtained from the filamentation in acetone. **b** Polarization measurement of terahertz radiation from air (red square) and acetone (blue circle) at a laser energy of 10 mJ. The theoretical cosine square fits to the plots are shown by red dashed line for air, and blue dashed line for acetone. The polarization of the incident laser is shown by a black dash-dot arrow. The polarization of the terahertz radiation from air (two-color) is shown by a red dash-dot arrow, which is rotated by 151° with respect to the incident laser polarization. The terahertz polarization for acetone (single color) is shown by a blue dash-dot arrow, rotated by 84° with respect to the laser polarization

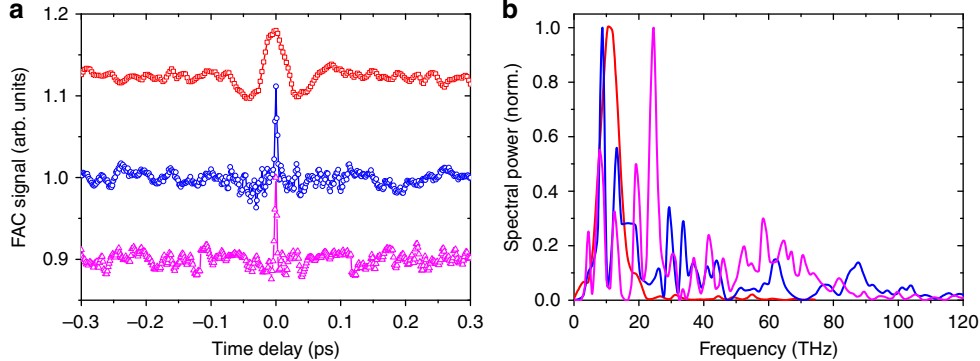

**Fig. 3** Broadband terahertz spectral measurements with Michelson interferometer. **a** Time domain FAC traces for terahertz radiation from two-color filamentation of air (red squares), and single-color filamentation of acetone (blue circles) and ethanol (magenta up-triangle) at an input laser energy of 10 mJ. The full-width at half-maxima for air is ~86 fs, while that for acetone and ethanol is ~19 fs. **b** Corresponding frequency domain power spectra obtained using fast Fourier transforms for air (red line), acetone (blue line), and ethanol (magenta line). For air, 99.5% of the spectral power lies between 0.1–50 THz. Acetone and ethanol, respectively, have 72 and 64% of the spectral power in the same frequency range

to be rotated by almost 90° with respect to the incident 800 nm pulse polarization for both air and acetone. This is consistent with the observations of Kosareva et al.[29], where the polarization of an independently launched second harmonic probe propagating through a filament was rotated due to cross-phase modulation. Since, the polarization of generated terahertz follows the polarization of second harmonic[30], it may be hypothesized that the generated broadband terahertz would be significantly depolarized with the polarization axis perpendicular to the incident laser polarization in agreement with our experimental observations.

**Field autocorrelation measurements**. To determine the spectral content of the broadband terahertz generated during the filamentation in liquids, the PED is removed from $F_1$, and the terahertz radiation is guided to a Michelson interferometer field autocorrelation (FAC) setup[9] (Fig. 1). The FAC signal is detected by the Gentec PED located at $F_2$. This scheme of measurement is independent of dispersion and absorption properties of crystals

associated with electro-optic sampling, and is ideal for measuring broadband spectra. Figure 3a shows typical FAC traces obtained from filamentation in air, acetone, and ethanol for $U_L$ ~10 mJ. Due to the low repetition rate of the laser (10 Hz) and turbulence generated in the liquids during filamentation, the FAC measurements are noisy. Therefore, to improve the signal to noise ratio, each FAC trace is averaged over three consecutive scans, where each data point in a scan is an average of 16 acquisitions. The full-width at half-maximum ($T_{FWHM}$) for acetone and ethanol is ~19 fs, which is about 4.5 times smaller than the air $T_{FWHM}$ (~86 fs), indicating that the liquid terahertz spectrum has a larger bandwidth (see Methods).

Figure 3b shows the broadband terahertz power spectra corresponding to the FAC signal. The spectral power falls off at ~20 THz for air, and at ~100 THz for acetone and ethanol. For air, 99.5% of the spectral power lies between 0.1 and 50 THz. In comparison, acetone has 72% of the spectral power in the same range. For ethanol, the number is 64%. The spectral profile shows several peaks and troughs, which can be attributed to the absorption in plasma, the filters, and atmospheric water vapor.

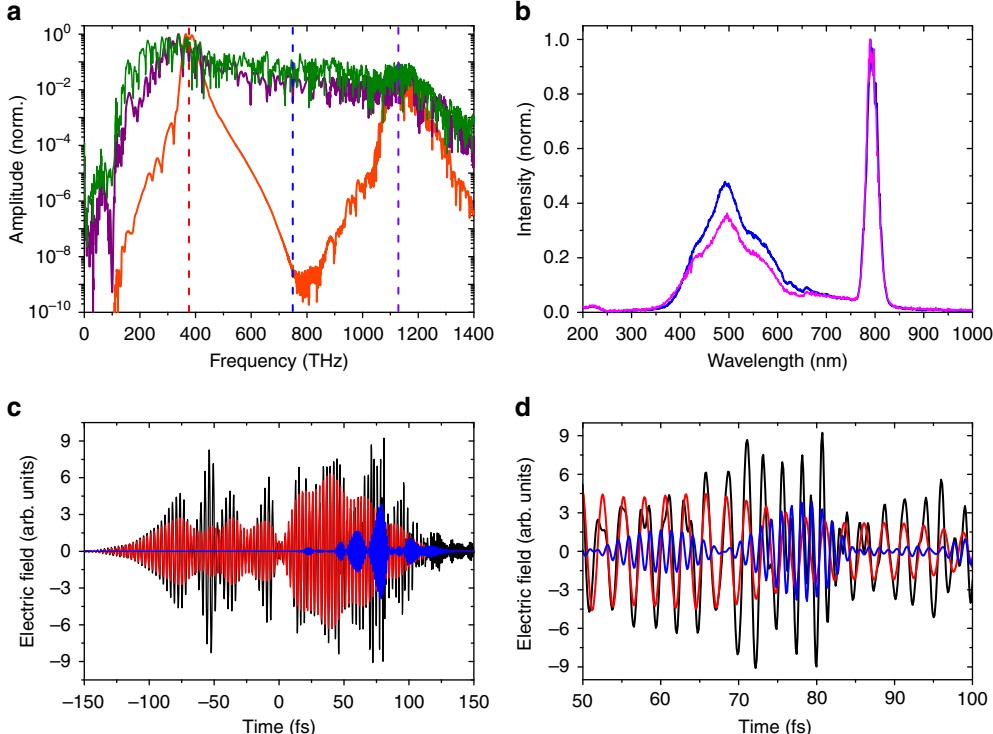

**Fig. 4** Simulation of the broadband terahertz generation mechanism and optical spectrum measurements. **a** Simulated laser-induced supercontinuum spectra obtained from the simultaneous solution of the UPPE with the kinetic equation for the concentration of free electrons in ethanol at laser input energies of 0.0674 μJ (orange line), 0.674 μJ (purple line), and 6.74 μJ (green line). Terahertz radiation is shown to be generated at higher energies (0.674, 6.74 μJ) by the two-color mechanism, where the second harmonic is a part of the supercontinuum. The vertical dashed lines shows the fundamental (800 nm, red), second harmonic (400 nm, blue), and the third harmonic (266 nm, violet) on the frequency scale. **b** Measurement of UV to infrared spectrum emanated during single-color filamentation of ethanol (magenta) and acetone (blue) demonstrating different ratios of second harmonic generation with respect to the fundamental. **c** Simulation of the temporal profile of the laser electric field inside the liquid at $z = 25$ mm. The full electric field ($E_{full}$) is shown by black lines, the electric fields corresponding to the fundamental ($E_\omega$) and second harmonic ($E_{2\omega}$) frequencies are shown by red and blue lines, respectively. **d** The simulated laser electric fields $E_{full}$ (black line), $E_\omega$ (red line), and $E_{2\omega}$ (blue line) with expanded time scale between 50–100 fs to show details of the phase variations

The unbalanced reflection/transmission characteristics of the pellicle beam-splitter also contribute to the spectral profile. It is observed that the liquid spectrum has negligible spectral power below 4 THz. This is independently confirmed for all the liquids used in the experiments (Fig. 2a) by noting that there is no terahertz transmission through a low-pass black polyethylene filter with sharp cut-off near 3 THz. On increasing the laser energy for a given liquid, the turbulence and bubbling in the liquid increases, and leads to noisy interferogram (low signal to noise ratio). With different liquids, there are qualitative differences in the spectrum, although the overall bandwidth remains almost the same. On average, 70% of the spectral power is in the 0.1–50 THz range for the liquids.

**Nonlinear pulse propagation simulations**. In order to elucidate the underlying physics of the broadband terahertz generation in liquids, we performed a series of simulations based on the uni-directional pulse propagation equation (UPPE)[31, 32] (see Methods). The simulations are performed for ethanol since its dispersive properties are well studied across the whole spectrum[26, 33]. Figure 4a shows the simulated amplitude spectra (integrated over the beam cross-section) for laser pulses with three different peak powers (energies): 1.2 $P_{cr}$ (0.0674 μJ), 12 $P_{cr}$ (0.674 μJ), and 120 $P_{cr}$ (6.74 μJ), where $P_{cr} = 0.71$ MW is the critical power for self-focusing in ethanol. The simulated spectra are obtained at a propagation distance of 28 mm from the

entrance of the cuvette, right after the end of the filamentation region (in the simulations). In the case of 1.2 $P_{cr}$ the pulse spectrum exhibits spectral broadening around the fundamental and the third harmonic but does not reach the terahertz range. However, with the increase of the pulse peak power up to 12 $P_{cr}$, a very broad pulse spectrum (i.e., supercontinuum) starts to form, ranging from ultraviolet to terahertz frequencies. A further increase of the pulse peak power, up to 120 $P_{cr}$, results in an even stronger supercontinuum with more energy in the terahertz range. Therefore, at high peak powers the broad spectrum of the pulse becomes almost flat over the frequencies between the fundamental and third harmonic, with efficient energy transfer to the frequencies around the pulse second harmonic due to spectral broadening. The second harmonic, being mixed with the fundamental, generates the terahertz part of the spectrum by the four-wave mixing and the asymmetric plasma current mechanism similar to the two-color filamentation in air, where one injects the second harmonic pulse separately[9].

To verify the suggested physical explanation from the simulations, we recorded the visible part of the supercontinuum from filamentation in ethanol and acetone ($U_L \sim 10$ mJ), shown in Fig. 4b. The measurements are performed at the rear wall (exit) of the cuvette, with the collecting optic fiber oriented perpendicular to the direction of laser propagation to sample the optical radiations scattered by the cuvette wall. This is done to sample the supercontinuum without using any filters to suppress the 800 nm fundamental. The spectrum depicts the existence of a strong

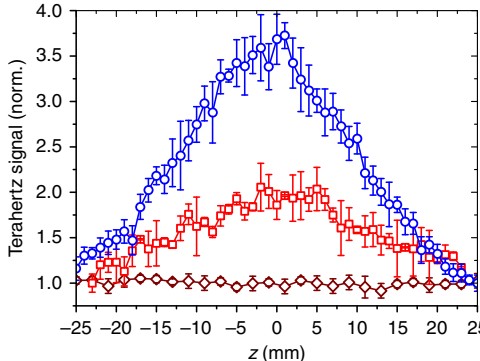

**Fig. 5** Nonlinear absorption in an open-aperture z-scan in doped silicon. Transmitted broadband terahertz signal (normalized with respect to the minimum) through LR-Si as a function of the distance along the axis of the focus $F_1$ ($z = 0$ is the focus) for terahertz radiation generated from two-color filamentation of air (red squares), and single-color filamentation of acetone (blue circles) at 10 mJ. Terahertz signal transmitted through HRFZ-Si (dark-red diamonds) as a function of z is shown for reference. Each data point is averaged over 16 acquisitions and the average and standard deviation are recorded. The error bars are obtained using the relation: error in the normalized scale = average value normalized to the minimum × standard deviation/average

supercontinuum around the second harmonic, in agreement with the simulations. Note that the spectral intensity at wavelengths below 350 nm is small because the ultraviolet part of the spectrum is strongly absorbed in the liquids. One can also observe that the second harmonic signal is higher for acetone compared to ethanol, indicating higher terahertz yield, which agrees with the terahertz output trend in our experiments (Fig. 2a).

To support the proposed mechanism of terahertz generation, we take the full field of laser pulse inside the filamentation region and separate the parts that correspond to the fundamental and second harmonic. We calculate the pulse spectrum and apply to it two Gaussian-shaped filters that are centered at frequencies of the fundamental and second harmonic and whose widths are equal to 10% of the corresponding frequencies. Figure 4c shows the field of the laser pulse ($E_{full}$) at 12 $P_{cr}$ and at a distance $z = 25$ mm. On the same figure we plot the fundamental ($E_\omega$) and second harmonic ($E_{2\omega}$) parts of this field as well. The field variations are replotted with an expanded time scale (50–100 fs) in Fig. 4d. We see that the fields of the fundamental and the second harmonic fully overlap in time and their amplitudes have comparable magnitudes. The appearance of such strong two-color fields can justify the high efficiency of terahertz generation in our experiments. This two-color scheme is found to be extremely efficient here since the two waves are in phase along the whole filamentation region. This happens because the second harmonic is generated during the propagation, something that is fundamentally different from the two-color scheme in gases, where the two waves are independent and would rapidly walk-off in the liquids because of their strong dispersion.

Due to computational limitations, we can simulate only pulses with energy of several μJ, while in the experiments we use pulse energies up to 28 mJ. Nevertheless, by extrapolating the results of the simulations, the appearance of the strong terahertz signal under experimental conditions can be explained.

**Nonlinear terahertz transmission**. To illustrate the nonlinear optical effect using the broadband terahertz pulses obtained from filamentation in liquids, an open aperture z-scan is performed on a 0.5 mm thick low-resistivity silicon (LR-Si) having electron

concentration of $\sim 10^{15}$ cm$^{-3}$. Figure 5 shows the normalized terahertz signal transmitted through LR-Si, as it is scanned along the terahertz focus $F_1$ (Fig. 1). For $U_L \sim 10$ mJ, the terahertz energy at $F_1$ for air and acetone are ~1 and 37 μJ, respectively, which gives electric field amplitudes ($E_{THz}$) of ~0.3 MV cm$^{-1}$ (air) and ~2 MV cm$^{-1}$ (acetone) for a terahertz focal diameter of ~1 mm (measured by knife edge scanning), assuming 1 ps pulse duration. A z-scan performed on high-resistivity float-zone silicon (HRFZ-Si) with $E_{THz}$ ~2 MV cm$^{-1}$ is plotted on the same graph (Fig. 5) to demonstrate that the peaks observed for LR-Si are not from some geometrical effect. The scans demonstrate the typical transmission enhancement associated with the phenomenon of inter-valley scattering of electrons in the conduction band, which scales with increasing terahertz electric field[34, 35].

## Discussion

In conclusion, we have demonstrated an unconventional way of generating high-energy, ultra-broadband terahertz pulses by ultrafast laser filamentation in liquids, obtaining a remarkably high conversion efficiency >10$^{-3}$, without the use of any crystal[13] or high-vacuum[16]. Our simulations show that the efficient generation of terahertz radiation in liquids can be explained by the local in-phase generation of a strong second harmonic component as part of the nonlinear spectral broadening. Further studies exploring the mechanisms for manipulation of the spectrum are underway.

The exact mechanism of broadband terahertz transmission through the absorbing liquid column requires further studies for better understanding. From our simulations, that take into account all linear and nonlinear effects, it is evident that the key parameter for observing significant terahertz energy after the cuvette is to have a filament that occupies the whole length of it and most importantly that the filament reaches close to the rear cuvette wall to avoid further linear absorption losses.

This method of broadband terahertz generation from liquids is expected to find wide use in nonlinear terahertz optics and spectroscopy, and would be of immense interest to researchers spanning various disciplines.

## Methods

**Filter transmission**. The filter transmission characteristics shown in Supplementary Fig. 1a are obtained using a Fourier transform infrared (FTIR) spectrometer (JASCO FT/IR-4100A). It is observed that mid-IR radiation ($\lambda$ ~1–5 μm) generated during filamentation in the liquids is transmitted through the standard HRFZ-Si filters along with the terahertz radiation. Supplementary Fig. 1b shows the resultant transmission for various filter combinations. The filter combination of two HRFZ-Si and one black-low-resistivity silicon of 0.5 mm thickness each, and one 5 mm thick high-density polyethylene (HDPE) is placed before the PED to minimize the contribution of the mid-IR region to the integrated terahertz energy and FAC measurements. This filter combination has a transmission of less than 10$^{-4}$ up to ~15 μm (~20 THz). The transmission rises above ~10$^{-2}$ at about 23 μm (~13 THz). In the terahertz region (beyond the range of the standard FTIR spectrometer used in these measurements), the equivalent transmission for this combination is ~0.1 on average. Thus, the attenuation of frequencies above 20 THz by the filters is about three orders of magnitude higher compared to that in the terahertz regime. Therefore, the filter combination used in this experiment significantly attenuates the mid-IR frequencies compared to the terahertz frequencies. Thicker HDPE filters are used at higher energies and a corresponding transmission correction is applied to obtain integrated measurements.

**Calculation of integrated energy**. The integrated energy is calculated from the knowledge of the spectral power in the 0.1–50 THz range and the transmittance of the filters in the following manner. The integrated radiation collected by the calibrated PED in the presence of the filter combination is recorded for example as, $x$ μJ. Since on an average ~70% of the detected radiation lies in the 0.1–50 THz range for the liquids, the integrated broadband terahertz energy in this range is estimated as $y = 0.7x$ μJ. Now we apply the correction for the filter combination, which is conservatively taken as ~0.1 over the whole 0.1–50 THz range, and obtain the terahertz energy as $U = y/0.1 = 7x$ μJ.

Thus $U$ (in μJ) is the estimated integrated broadband terahertz energy radiated from the liquid source due to femtosecond laser filamentation.

**Cuvette length**. The length of the cuvette is chosen so that there is negligible filamentation and supercontinuum generation at the entrance wall (wall thickness ~0.5 cm) due to the laser, while ensuring that it is not too long to attenuate the terahertz significantly after generation. The inner wall-to-wall cuvette length of 5 cm is optimum for the 50 cm focal length lens used in this experiment. For longer focal lengths, longer cuvettes must be used.

**Optical polarization**. The optical polarization of the fundamental (800 nm) laser pulse, and the second harmonic (400 nm) pulse generated (part of the super-continuum) after filamentation in air (without the BBO) and acetone are shown in Supplementary Fig. 2. The polarization plots were obtained using standard sheet polarizers, photo-diodes, and narrow-band filters centered around 800 and 400 nm, respectively. The polarizer was placed between $OAP_1$ and $OAP_2$, and the photo-diode with the filters was at $F_1$ (Fig. 1). From Supplementary Fig. 2 we can see that the 800 nm pulse gets depolarized post filamentation, and the degree of depolarization is more in case of liquids. The generated 400 nm is orthogonally polarized with respect to the initial polarization of the 800 nm (before filamentation), both in the case of air and liquids. In case of terahertz from air, the supercontinuum generated second harmonic has a limited role, since the externally injected (via the BBO) 400 nm is dominant, and controls the polarization of the terahertz radiation. In the case of liquids, the supercontinuum generated 400 nm is responsible for generation of the terahertz radiation, and hence the terahertz polarization is found to be orthogonal with respect to incident 800 nm. In both cases, since the 800 nm pulse suffers depolarization, the contrast between the maxima and minima is poor for the measured terahertz polarization (Fig. 2b).

**Terahertz beam profile**. Supplementary Fig. 3 shows a comparison between the 400 nm emission after the filamentation in acetone (in 5 cm cuvette) and the terahertz beam profile. The terahertz beam profile is approximately a Gaussian with an FWHM of ~11 mm, while the 400 nm has an FWHM of ~10 mm.

**Fitting of FAC traces**. The FAC traces are fitted with a function of the form $A(\tau) = y_o + A_o \cos(\omega\tau)\exp\{-2(\tau/T_o)^2\}$, where $\tau$ is the time delay, $A(\tau)$ is the FAC signal, $y_o$ is the offset, $A_o$ the amplitude, $\omega$ is the fundamental frequency, and $T_o = T_{FWHM}/\sqrt{2\ln 2}$, $T_{FWHM}$ being the time delay width at half of the signal maximum.

**Simulations**. The UPPE[31, 32] can be written as:

$$\frac{\partial \hat{E}}{\partial z} = ik_z\hat{E} + i\frac{\mu_o\omega^2}{2k_z}\left[\hat{P}_{nl} + \frac{i}{\omega}\left(\hat{J}_f + \hat{J}_a\right)\right] \quad (1)$$

where $\hat{E}(k_x, k_y, k_z)$ is the spatio-temporal spectrum of the laser pulse, $k_z(k_x, k_y, \omega) = [k^2(\omega) - k_x^2 - k_y^2]^{1/2}$ is the propagation constant with $k_x$, $k_y$, and $\omega$ being the spatial and temporal angular frequencies, $k(\omega) = n(\omega)\omega/c_o$ is the wave number, $n(\omega)$ is the frequency-dependent refractive index, $c_o$ is the speed of light in vacuum, and $\mu_o$ is the vacuum permeability. The nonlinear term in Eq. (1) includes the third-order nonlinear polarization, $\hat{P}_{nl} = \varepsilon_o\chi^{(3)}\hat{E}^3$, the current of free electrons $\hat{J}_f = (q_e^2/m_e)[(\nu_c + i\omega)/(\nu_c^2 + \omega^2)]\hat{\rho}\hat{E}$, and the current that is responsible for nonlinear absorption, $\hat{J}_a = K\hbar\omega_o\left(\frac{1}{E}\frac{\partial\rho}{\partial t}\right)$. Here the ∧ denotes the spatio-temporal spectrum, $\varepsilon_o$ is the vacuum permittivity, $\chi^{(3)} = 4n_o^2\varepsilon_o c_o n_2/3$ is the cubic susceptibility with $n_2$ being the nonlinear index, $n_o$ is the medium refractive index at the pulse central frequency $\omega_o$, $q_e$ and $m_e$ are the charge and mass of the electron, $\nu_c$ is the collision frequency, $\rho$ is the concentration of free electrons (m$^{-3}$), while $K = \text{Int}\{U_i/\hbar\omega_o\} + 1$ with Int denoting the integer part, and $U_i$ being the band gap of the liquid.

Together with the UPPE we solve the kinetic equation for the concentration of free electrons:

$$\frac{\partial\rho}{\partial t} = R_1(\rho_{nt} - \rho) + R_2\rho \quad (2)$$

where $\rho_{nt}$ is the concentration of neutral atoms, with $R_1$ and $R_2$ being the optical field and avalanche ionization rates. For $R_1$ we use the Keldysh formula, while $R_2$ is given by $R_2 = \sigma(\omega_o)E^2/U_i$, where $\sigma(\omega_o) = (q_e^2/m_e)[\nu_c/(\nu_c^2 + \omega_o^2)]$ is the inverse Bremsstrahlung cross-section. For the calculation of $\partial\rho/\partial t$ in the expression for $\hat{J}_a$ we use the first term on the right-hand side of Eq. (2).

As the initial condition for Eq. (1) we take a Gaussian pulse at the entrance of the cuvette with liquid inside. The parameters of the pulse are similar to the experimental ones: 800 nm central wavelength, 48 fs FWHM pulse duration, 1.53 mm FWHM beam size, and 25 mm focal distance. The plasma density is ~6 × 10$^{21}$ cm$^{-3}$ for 6.74 μJ pulses.

Using the above model and input condition, we simulate the laser pulse propagation inside the cuvette filled with ethanol. We choose ethanol, because, compared to other liquids in our experiments, its dispersive properties in the far-infrared frequency range are studied the most in the literature. In our simulations we use the following ethanol parameters: the refractive index $n(\omega)$ is calculated using the data from ref. [26], $n_2 = 10^{-20}$ m$^2$ W$^{-1}$ [33], $\nu_c = 1/3$ fs$^{-1}$ (we take the value for water[19, 36]), $U_i = 10.5$ eV[37], $\rho_{nt} = 1.03 × 10^{28}$ m$^{-3}$.

**Data availability**. The data corresponding to the plots presented in this study are available from the corresponding author on request.

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

## Acknowledgements

G.R.K. acknowledges partial support from J.C. Bose Fellowship grant JCB-037/2010 (Department of Science and Technology) and UGC-ISF grant F.6–15/2014 (IC) (University Grants Commission) of the Government of India. S.T., V.F., and A.D.K. acknowledge partial support from the National Priorities Research Programme grant number NPRP9-329-1-067 from the Qatar National Research Fund (member of The Qatar Foundation) and the European Union's Horizon 2020 Laserlab Europe (EC-GA 654148).

## Author contributions

G.R.K. and I.D. conceived the experiment. I.D. designed the details of the experiment in consultation with G.R.K., and was the main researcher to execute them in collaboration with K.J., A.D.K., A.M., M.S., D.S. and A.D.L. I.D. analyzed the data in consultation with G.R.K. and S.T. Simulations were carried out by V.F. in consultation with S.T. and A.C. The results were discussed by I.D., G.R.K., S.T., V.F., A.C. and A.D.K. The manuscript was prepared by I.D., G.R.K., S.T., V.F., A.D.K. and A.C. and discussed and approved by all the authors.

## Additional information

**Competing interests:** The authors declare no competing financial interests.

