## [Peer Review File · Nature Communications]

Reviewers' Comments:

Reviewer #1:

Remarks to the Author:

In the manuscript terahertz generation from single color ultrashort pulses in liquids is reported. The measured THz energy is about 20 times larger than what is obtained during two-color filamentation in air. The physical interpretation by second harmonic generation and ionization-induced plasma current generation similar as in two-color THz generation in gases is supported by simulations of the unidirectional propagation equation taking into account the optical Kerr effect and ionization-induced effects. In my knowledge the results are novel and will be of interest to others in the community. I propose the Publishing of this manuscript in Nature Communication.

Reviewer #2:

Remarks to the Author:

The paper by Indranuj Dey et al, demonstrates a THz power upscaling through ultrashort laser filamentation in liquids, obtaining ultra-broadband (~ 100 THz), highly energetic (~ 80 $\mu\text{J}/\text{pulse}$) THz pulses. Using single color filamentation in liquids, they generate THz radiation with energies that are an order of magnitude higher compared to two-color filamentation in gases. The results are supported by simulations showing formation of an asymmetric plasma current and thereby generation of THz radiation in a way similar to the two-color filamentation scheme in gases. The results presented in the manuscript are interesting and deserve a publication in a high-ranking journal. I recommend the manuscript for publication after a minor revision.

My main criticism concerns the following points:

1. Is the THz polarization from liquids independent of the liquid used? Can the degree of depolarization due to the liquids be estimated? It would be better to move some of the explanation of the polarization data, from the Methods section to the main text.
2. The authors say that both 4-wave mixing and 2-color filamentation are responsible for THz generation from the liquids. Which has the higher contribution, or matches well with experimental data?
3. What are the typical plasma densities achieved in the liquid? What plasma densities are used in the model?
4. What is the minimum laser energy at which the THz signal has been detected from the liquids?
5. Does the THz spectrum vary with laser energy or the type of liquid used? Is it detectable experimentally?
6. What is the effect of pulse-width on the THz generation? Has it been measured?
7. Is it possible to obtain the far-field spectrum and polarization of the THz from the simulation?
8. In Fig. 4c, both the optical spectrum for Acetone and Ethanol show a peak at $\sim 530 - 550$ nm. Interestingly, no such peak is predicted by the model. What is the possible reason for these peaks?

Reviewer #3:

Remarks to the Author:

The manuscript describes high-energy terahertz (THz) generation from laser filamentation in liquids. In particular, it reports several tens of microjoules of THz radiation energy with laser-to-THz conversion efficiency of $>0.1\%$, which is an order of magnitude higher than that in typical two-color filamentation in gases. The authors also claim that their laser-produced filaments in liquid can generate extremely broadband (>100 THz) radiation. I feel that THz generation from filamentation in liquids is novel and could be an interesting subject in the community.

The main concern, however, is that the measured radiation mostly falls into the mid-IR frequency (>60 THz or <5 microns) having little or no radiation below 50 THz. This is evident from the field autocorrelation trace shown in Fig. 3(a), which is quite noisy and not symmetric at all outside (-0.05 ps, 0.05 ps). The oscillatory noise seems to produce low frequency radiation below 50 THz. For comparison, the air signal in Fig. 3(a) is obvious around (-0.1 ps, 0.1 ps), and this produces strong THz radiation at 0-20 THz. However, this amount of "oscillation period" and "amplitude" is not shown in the acetone trace even though the Fourier-transformed spectrum shows strong THz radiation at 0-20 THz. In addition, the provided simulation shown in Fig. 4(a) supports new frequency generation from the high frequency side, with dramatically falling radiation power as the frequency decreases. The simulation indeed justifies supercontinuum generation over a broad range of spectrum, but not exclusive THz generation at low frequencies (<50 THz).

Given these, I believe that most of energy that the authors measured, in fact, is mid-IR, not covering the conventional THz frequency band at 0.1-10 THz. Note that a large amount of mid-IR radiation is cut down by the HDPE filter used prior to detection (see Fig. 1 in the supplementary materials). This means that mid-IR radiation is dominantly produced from filaments in liquids.

If the authors believe that their source indeed produces 0-50 THz radiation, then the authors must provide more solid evidence such as "repeatable" autocorrelation traces possibly for more liquids. Currently, the claim of ~100 THz bandwidth relies on only "one scan", and this scan is not trustworthy because of its seemingly large amount of noise.

Another issue is that even if the slow oscillations outside (-0.05 ps, 0.05 ps) is real, those oscillations appear to spread out over the entire time window on (-0.3 ps, 0.3 ps). This indicates that the radiation is not ultrafast, and it may originate from thermal black-body radiation, not necessarily from two-color mixing in filamentation.

As the authors note, strong THz generation from polar liquids is counterintuitive because of strong THz absorption. For instance, the absorption (or attenuation) constant of deionized water is $100\text{-}1000\text{ cm}^{-1}$ at 1-100 THz (P. H. Siegel, IEEE Trans. Microwave Theory Techniques 52, 2438 (2004)). This means that 1-100 THz radiation cannot literally propagate even over 1 cm, one fifth of the cuvette length. So the measured energy in Fig. 2 for DI-water must correspond to frequencies at >100 THz. Even in ethanol, transmission drops significantly below 20 THz. Therefore the authors need to justify their claims by estimating the exact amount of THz absorption in each liquid. In addition, it is not clear if the provided simulation takes into account linear THz absorption at a broad range of frequency for ethanol and/or other liquids.

Also, the simulation provided in the manuscript does not provide realistic experimental conditions. The laser energy assumed in the simulation is several microjoules only, whereas the real experiments are performed at a few tens of millijoules, about 2~3 orders of magnitude larger. This big discrepancy makes it hard to justify the exact mechanism of THz radiation at 0-50 THz, let alone to predict any resulting THz radiation spectra.

It is not clearly explained why the polarization of the emitted THz radiation (and the second harmonic) is perpendicular to the input polarization. The provided references (Ref. 22 and Kosareva et al. in the Methods part) do not explain either.

The followings are minor questions/comments.

In Fig. 4(c), the 400 nm signal looks quite strong compared to the 800 nm peak. Is there any filter used to attenuate 800 nm? If so, it should be clarified.

What does "HRFZ-Si" stand for?

Response to the Reviewers

Reviewer #1

Comments:

“In the manuscript terahertz generation from single color ultrashort pulses in liquids is reported. The measured THz energy is about 20 times larger than what is obtained during two-color filamentation in air. The physical interpretation by second harmonic generation and ionization-induced plasma current generation similar as in two-color THz generation in gases is supported by simulations of the unidirectional propagation equation taking into account the optical Kerr effect and ionization-induced effects. In my knowledge the results are novel and will be of interest to others in the community. I propose the Publishing of this manuscript in Nature Communication.”

Response: Thank you very much for your encouraging comments and appreciation of our results, and recommending publication in Nature Communications.

Reviewer #2

Initial Comments:

“The paper by Indranuj Dey et al, demonstrates a THz power upscaling through ultrashort laser filamentation in liquids, obtaining ultrabroadband (~ 100 THz), highly energetic (~ 80 $\mu\text{J}/\text{pulse}$) THz pulses. Using single color filamentation in liquids, they generate THz radiation with energies that are an order of magnitude higher compared to two-color filamentation in gases. The results are supported by simulations showing formation of an asymmetric plasma current and thereby generation of THz radiation in a way similar to the two-color filamentation scheme in gases. The results presented in the manuscript are interesting and deserve a publication in a high-ranking journal. I recommend the manuscript for publication after a minor revision.”

Response: Thank you very much for your encouraging comments on the significance of our paper and recommending publication in Nature Communications. We address your specific concerns below.

Specifics:

1. “Is the THz polarization from liquids independent of the liquid used? Can the degree of depolarization due to the liquids be estimated? It would be better to move some of the explanation of the polarization data, from the Methods section to the main text.”

Response: The THz polarization from the liquids (acetone, methanol, ethanol, 1,2-dichloroethane) is $\sim 90^\circ \pm 10^\circ$ with respect to the input laser polarization.

A first order estimation of the degree of depolarization can be done by comparing the ratio of the polarization maxima to the minimum (contrast). The 800 nm optical input has a contrast of ~ 8.9 , while the 800 nm laser emerging after filamentation (input energy ~ 10 mJ) in 5 cm of acetone has a contrast ratio of ~ 1.1 (implying that it is almost circular). So, the laser is $\sim 88\%$ depolarized after filamentation with respect to the input polarization. In comparison, the 800 nm contrast after filamentation in air is ~ 2.1 , i.e., $\sim 77\%$ depolarization.

The contrast of the THz polarizations for both air and acetone (in 5 cm cuvette) is ~ 1.2 as seen in Fig. 2b.

Modifications in the paper:

Relevant portions of the explanation of the polarization data has been moved to the main text from the Methods section. We have also extended the discussion on the origin of these polarization phenomena as per the reply to point 4 raised by the 3rd reviewer (see below).

2. “The authors say that both 4-wave mixing and 2-color filamentation are responsible for THz generation from the liquids. Which has the higher contribution, or matches well with experimental data?”

Response: Since filamentation itself is the result of dynamical balance between the Kerr self-focusing and plasma defocusing, there is no explicit experimental way to separate 4-wave mixing and photocurrent contributions to THz generation. Recent theoretical studies of two-color filamentation in gases [Bergé, Phys. Rev. Lett. **110**, 73901 (2013); Borodin, Opt. Lett. **38**, 1906 (2013); Andreeva, Phys. Rev. Lett. **116**, 063902 (2016)] show that the contribution of 4-wave mixing is much weaker and generates higher THz frequencies than 2-color

filamentation (photocurrent mechanism). In turn, the relative contribution of 4-wave mixing and photocurrent mechanism in liquids is still unexplored (which is not surprising since our study is the first evidence that such THz generation is possible in principle). However, one can expect that the role of 4-wave mixing in liquids is much higher than in gases due to orders of magnitude higher nonlinear indices. Detailed analysis of the interplay between the above two mechanisms in condensed media is the theme of further studies.

3. “What are the typical plasma densities achieved in the liquid? What plasma densities are used in the model?”

Response: The typical plasma densities in liquids are of the order of few times 10^{21} cm^{-3} [Minardi, Appl. Phys. Lett. **105**, 224104 (2014)]. In simulations with 6.74 μJ pulses we obtained peak plasma density up to $\sim 6 \times 10^{21} \text{ cm}^{-3}$.

Modifications in the paper:

We added the corresponding values to the main text.

4. “What is the minimum laser energy at which the THz signal has been detected from the liquids?”

Response: The minimum energy measurable (above noise level) by the Gentec THz detector is $\sim 25 \text{ nJ}$. This limits the minimum laser energy $\sim 0.1 \text{ mJ}$ for acetone (with the Black-Si filter + 5 mm HDPE + 0.5 mm HRFZ-Si present before the detector).

Modifications in the paper:

We have incorporated the above information in the manuscript.

5. “Does the THz spectrum vary with laser energy or the type of liquid used? Is it detectable experimentally?”

Response: On increasing the laser energy for a given liquid, the turbulence and bubbling in the liquid increases, and leads to a noisy interferogram (low signal to noise ratio). With different liquids, there is qualitative differences in the spectrum, though overall the bandwidth remains almost the same.

Modifications in the paper:

We have added more interferograms in the experimental data of the main text, and also added the above comments for clarification.

6. “What is the effect of pulse-width on the THz generation? Has it been measured?”

Response: The effect of pulse-width on the THz generation from the liquids has been measured by varying the inter-grating distance in the compressor. It has been observed that the maximum THz output is obtained close to (but not exactly equal to) the optimum grating position (and pulse-width) for air THz generation ($\sim 50 \text{ fs}$). The THz output falls off drastically at longer

pulse-width. Experiments are in progress in this regard, and the results would be reported in the near future.

7. “Is it possible to obtain the far-field spectrum and polarization of the THz from the simulation?”

Response: The spectra that we provide in the main text are the far-field spectra, since in order to calculate them we use an angular filter that cuts all waves that propagate under large angles, including evanescent waves.

In our simulations we assume that the polarization of the pulse and all generated frequencies is linear and does not change during the propagation.

8. “In Fig. 4c, both the optical spectrum for Acetone and Ethanol show a peak at ~ 530 – 550 nm. Interestingly, no such peak is predicted by the model. What is the possible reason for these peaks?”

Response: The peak at ~ 540 nm seemed quite interesting, so we repeated the experiment, but were unable to reproduce them. We believe that they were artifact of the measurement due to some preferential scattering of the green part of the supercontinuum into the optical fiber.

Modifications in the paper:

We have updated the plot in Fig. 4 (c) with the revised optical spectrum data.

Reviewer #3

Initial Comments: “The manuscript describes high-energy terahertz (THz) generation from laser filamentation in liquids. In particular, it reports several tens of microjoules of THz radiation energy with laser-to-THz conversion efficiency of $>0.1\%$, which is an order of magnitude higher than that in typical two-color filamentation in gases. The authors also claim that their laser-produced filaments in liquid can generate extremely broadband (>100 THz) radiation. I feel that THz generation from filamentation in liquids is novel and could be an interesting subject in the community.”

Response: Thank you very much for your positive comments on our paper and its appropriateness for Nature Communications. We address your specific concerns below.

Specifics:

1. “The main concern, however, is that the measured radiation mostly falls into the mid-IR frequency (>60 THz or <5 microns) having little or no radiation below 50 THz. This is evident from the field autocorrelation trace shown in Fig. 3(a), which is quite noisy and not symmetric at all outside (0.05 ps, 0.05 ps). The oscillatory noise seems to produce low frequency radiation below 50 THz. For comparison, the air signal in Fig. 3(a) is obvious around (0.1 ps, 0.1 ps), and this produces strong THz radiation at 0-20 THz. However, this amount of “oscillation period” and “amplitude” is not shown in the acetone trace even though the Fourier transformed spectrum shows strong THz radiation at 0-20 THz. In addition, the provided simulation shown in Fig. 4(a) supports new frequency generation from the high frequency side, with dramatically falling radiation power as the frequency decreases. The simulation indeed justifies supercontinuum generation over a broad range of spectrum, but not exclusive THz generation at low frequencies (<50 THz).

Given these, I believe that most of energy that the authors measured, in fact, is mid-IR, not covering the conventional THz frequency band at 0.1-10 THz. Note that a large amount of mid-IR radiation is cut down by the HDPE filter used prior to detection (see Fig. 1 in the supplementary materials). This means that mid-IR radiation is dominantly produced from filaments in liquids. If the authors believe that their source indeed produces 0-50 THz radiation, then the authors must provide more solid evidence such as “repeatable” autocorrelation traces possibly for more liquids. Currently, the claim of ~ 100 THz bandwidth relies on only “one scan”, and this scan is not trustworthy because of its seemingly large amount of noise. Another issue is that even if the slow oscillations outside (-0.05 ps, 0.05 ps) is real, those oscillations appear to spread out over the entire time window on (-0.3 ps, 0.3 ps). This indicates that the radiation is not ultrafast, and it may originate from thermal blackbody radiation, not necessarily from two-color mixing in filamentation.”

Response: We address the concerns in three parts.

(a) Filtering of the mid-IR: As already mentioned in the manuscript, filamentation in the liquids indeed could generate mid-IR radiation, and a combination of filters has been used to adequately attenuate the mid-IR radiation. The individual transmission of the different filters used, and the combined transmission for various filter combinations is shown below [plot (a) and plot (b) respectively]. The combination consists of a 0.5 mm thick black-low-resistivity silicon (Black-LR-Si), and a 5 mm high-density polyethylene (HDPE) along with two 0.5 mm thick high-resistivity-float-zone-Si (HRFZ-Si). The equivalent transmission of this combination is shown by the blue curve in the plot (b) below.

As can be seen, the transmission is below 10^{-4} , up to $\sim 15 \mu\text{m}$ ($\sim 20 \text{ THz}$). The transmission rises above $\sim 10^{-2}$ at about $23 \mu\text{m}$ ($\sim 13 \text{ THz}$). In the “THz” region (beyond the range of the standard FTIR spectrometer used in these measurements [JASCO FT/IR-4100A]), the equivalent transmission for this combination is ~ 0.1 on an average. Thus, the attenuation of frequencies above 20 THz by the filters is about 3 orders of magnitude higher compared to that in the THz regime. Therefore, we believe that the filter combination used in this experiment significantly attenuates the mid-IR frequencies compared to the THz frequencies. What we measure is essentially THz radiation.

To further elucidate the point, we present below a table showing the integrated measurement of the radiation by the Gentec detector, with various combinations of filters shown above. If we assume that the radiation is only in the mid-IR regime ($< 5 \mu\text{m}$), then applying the filter correction appropriately for each of the combinations should give almost the same integrated value. However, as can be seen from the data below, the corrected values for different combinations is quite different from that with only $2 \times \text{HRFZ-Si}$ (high transparency in the mid-IR regime) filter. Thus, the measured radiation has large contribution from frequencies below 50 THz and the mid-IR frequencies are significantly attenuated.

Filter	Avg. Transmission (mid-IR)	Measured signal (V)	Filter Corrected Signal (V)
$2 \times \text{HRFZ-Si}$ (0.5mm)	0.3	7.58	25.2
$2 \times \text{HRFZ-Si}$ (0.5mm) + Black-Si	5×10^{-3}	0.23	46.0
$2 \times \text{HRFZ-Si}$ (0.5mm) + Black-Si + HDPE (5mm)	10^{-4}	0.10	1000

Modifications in the paper:

The above clarifications and reasonings have been added to the revised Supplementary materials.

(b) Autocorrelation traces: The acetone autocorrelation trace reported in the manuscript is a typical trace. This is not a single scan, but an average of 3 consecutive scans. Furthermore, each data point in a scan is an average of 16 acquisitions.

We stress that the laser repetition frequency in our experiment is low (10 Hz), and a lot of turbulence is generated in the liquid during filamentation. These two factors make the measurements susceptible to noise and hence we averaged over several scans to improve the signal to noise ratio.

Evidently, we have autocorrelation traces for other liquids, and at different laser energies. We have included an extra trace for ethanol in the revised manuscript.

The asymmetry of the autocorrelation traces can be attributed to unequal splitting by the beam splitter which has different transmissions over the large bandwidth. This is similar to that observed in the case of THz from air by Kim et. al., *Nat. Phot.* **2**, 605 (2008).

(c) Black-body radiation: In addition to the measurements reported in the manuscript, we have also measured the variation of the integrated THz intensity with pulse-width of the laser. The THz output from the liquid decreases rapidly with increase of the pulse-width (since super-continuum generation efficiency decreases), in agreement with our modeling and proposed explanation with the in-phase second harmonic. The nature of the THz radiation is fully coherent as proven by all our measurements and the excitation studies inducing enhanced (nonlinear) transmission in doped Silicon via intervalley scattering. These data are in agreement to those reported recently by Kaur and Zhang [*IRMMW-THz 2010 - 35th International Conference on Infrared, Millimeter, and Terahertz Waves, Conference Guide* 10–11 (2010)]. The results demonstrate the importance of THz radiation from liquids for nonlinear THz studies.

2. “As the authors note, strong THz generation from polar liquids is counterintuitive because of strong THz absorption. For instance, the absorption (or attenuation) constant of deionized water is 100-1000 cm^{-1} at 1-100 THz (P. H. Siegel, *IEEE Trans. Microwave Theory Techniques* 52, 2438 (2004)). This means that 1-100 THz radiation cannot literally propagate even over 1 cm, one fifth of the cuvette length. So the measured energy in Fig. 2 for DI-water must correspond to frequencies at >100 THz. Even in ethanol, transmission drops significantly below 20 THz. Therefore, the authors need to justify their claims by estimating the exact amount of THz absorption in each liquid.”

Response: Indeed, liquids have large absorption in the THz region in general. However, the absorbance is not continuous throughout the spectral range and windows of low absorbance exist in between. Moreover, there are additional mechanisms at higher laser energies, in the filamentation regime (used in experiment). From our simulations, that take into account all linear and nonlinear effects, it is evident that the key parameter for observing significant THz energy after the cuvette is to have a filament that occupies the whole length of it and most importantly that the filament reaches close to the rear cuvette wall to avoid further linear absorption losses, which is what we have observed experimentally.

Modifications in the paper:

The above points are included in the discussion section of the revised manuscript.

3. “In addition, it is not clear if the provided simulation takes into account linear THz absorption at a broad range of frequency for ethanol and/or other liquids. Also, the simulation provided in the manuscript does not provide realistic experimental conditions. The laser energy assumed in the simulation is several microjoules only, whereas the real experiments are performed at a few tens of millijoules, about 2~3 orders of magnitude larger. This big discrepancy makes it hard to justify the exact mechanism of THz radiation at 0-50 THz, let alone to predict any resulting THz radiation spectra.”

Response: As the model of frequency-dependent complex refractive index of ethanol in our simulations we use the data from [Sani and Dell'Oro, *Optical Materials* **60**, 137 (2016)] (see the tables in their supplementary). These data are based on transmittance measurements, therefore they include the linear absorption. Since Sani and Dell'Oro measured the absorption of ethanol in the extremely wide spectral range, from 181 to $\sim 54000 \text{ cm}^{-1}$ (5.4 THz – 1.6 PHz or 0.19 μm – 55.25 μm), we are quite confident that in our simulations we properly take into account linear THz absorption for a broad frequency range.

In our numerical studies, we did not consider other liquids, precisely because for them we did not find such complete data on their optical properties.

Our simulations indeed cannot be used for direct quantitative comparison with the experiment. The simulations of the original problem with filamentation of multi-millijoule pulses in water will demand enormous computational resources and time and we doubt that anyone can perform them even using the most powerful cluster computers at the present time. However, there is no reason to doubt that during filamentation of high energy pulses the main physical mechanisms are still cubic Kerr and plasma nonlinearities that lead to broadening of the pulse spectrum and responsible for THz generation.

From this point of view our simulations with lower energy pulses, but still much above the critical power, can give us valuable information on the processes of THz generation in liquids. The key finding that explains the high energy conversion efficiency in the THz part of the spectrum is the strong second harmonic component that is generated in phase with the fundamental all along the filament – a fact confirmed by our experimental studies on the optical spectrum of the generated supercontinuum.

4. “It is not clearly explained why the polarization of the emitted THz radiation (and the second harmonic) is perpendicular to the input polarization. The provided references (Ref. 22 and Kosareva et al. in the Methods part) do not explain either.”

Response: Although this is a topic of further study we believe that it can be explained by the study of Yu et. al. [*Opt. Express* **18**, 12581 (2010)] that showed that any polarization perturbation induced by any optical element, like focusing lens or entrance window of the cuvette, is greatly magnified in the nonlinear propagation process. As a result, the generated supercontinuum becomes depolarized. Fig II in the supplementary material clearly shows this effect of depolarization of the supercontinuum.

Moreover, From Kosareva et. al. [*Opt. Lett.* **35**, 2904 (2010)] we know that the polarization of the second harmonic probe pulse co-propagating with the fundamental pump pulse during the filamentation in argon can be rotated by large angles, up to 90° . If the initial polarization of the second harmonic is not parallel to the polarization of the pump (as we expect in our case according to the above argument on depolarization) then its rotation is the most efficient.

The above two effects can explain why in our experiments the polarization of the second harmonic becomes rotated by large angles relative to the initial polarization (see Fig II in supplementary). Moreover, since the main mechanism responsible for polarization rotation is the cross-phase modulation, one can expect that rotation of the second harmonic polarization in liquids is much more efficient than in gases due to much stronger Kerr nonlinearities.

In turn, the polarization of generated THz follows the polarization of second harmonic [Esaulkov et. al., Front. Optoelectron. **8**, 73 (2014); Fedorov et. al., Plasma Phys. Control. Fusion **59**, 14025 (2017)].

Modifications in the paper:

The above clarifications have been incorporated in the revised text at the appropriate places.

Minor questions/comments:

Comment: “In Fig. 4(c), the 400 nm signal looks quite strong compared to the 800 nm peak. Is there any filter used to attenuate 800 nm? If so, it should be clarified.”

Response: The optical fiber of the spectrometer was placed near the cuvette, transverse to the propagation axis of the laser to collect the supercontinuum radiation scattered by the cuvette wall. This was done to minimize the influence of the 800 nm input without using any filters (which would limit the bandwidth of collection). That’s why the 800 nm peak looks comparable to the 400 nm peak. The position of the fiber optic was kept fixed for all the liquids.

Modifications in the paper:

The above clarification has been added to the revised manuscript.

Comment: “What does “HRFZ-Si” stand for?”

Response: HRFZ-Si stands for High Resistivity Float Zone Silicon.

Modifications in the paper:

This has been clarified in the revised text (Methods).

Reviewer #2 (Remarks to the Author):

The paper is ready for publication

Reviewer #3 (Remarks to the Author):

I feel nearly all referees' questions and concerns are adequately addressed. Also the manuscript improved a lot and can be published in Nature Communications if the following issues are resolved.

I still believe that the reported scheme produces a large amount of mid/long-wavelength infrared (IR) at >50 THz. In the response letter, the authors claim that what they measure is "essentially THz radiation" based on their filter choice. The fact that the combined filter transmission is less than 0.0001 above 20 THz whereas it is 0.01~0.1 below 20 THz doesn't mean that THz radiation above 20 THz is completely blocked. Apparently, the measured spectra shown in Fig. 3(b) confirm the presence of high-frequency THz radiation at 20-100 THz. Considering the strong attenuation factor (10^{-4}) of the combined filters, this scheme dominantly produces mid/long-wavelength IR prior to its detection! The measured autocorrelation traces also confirm strong mid/long-wavelength IR generation. The FWHM for acetone and ethanol is ~ 19 fs, and this corresponds to a radiation bandwidth (or peak radiation frequency) of ~ 50 THz.

Given that the conventional THz frequency band ranges from 0.1 to 10 THz, my suggestion is to call the observed radiation as THz/mid-IR radiation (or something equivalent), not simply THz radiation, throughout the paper.

One related question is how the maximal THz energy ~ 76 μ J was estimated? Does it represent the final energy measured by the pyroelectric detector? Or is the energy before the filter combination? If so, what factor was used to estimate the energy before the filters? Also this energy represents all radiation frequencies as shown in the spectrum figure?

Response to the Reviewers

Reviewer #2:

Comment:

The paper is ready for publication.

Response: Thank you for recommending the publication of our manuscript in Nature Communications. Your valuable comments have helped in improving and enriching our paper.

Reviewer #3:

Initial Comments:

I feel nearly all referees' questions and concerns are adequately addressed. Also the manuscript improved a lot and can be published in Nature Communications if the following issues are resolved.

Response: Thank you for appreciating our revisions and recommending publication of the manuscript in Nature Communications. We address your specific concerns below.

Specifics:

1. I still believe that the reported scheme produces a large amount of mid/long-wavelength infrared (IR) at >50 THz. In the response letter, the authors claim that what they measure is “essentially THz radiation” based on their filter choice. The fact that the combined filter transmission is less than 0.0001 above 20 THz whereas it is 0.01~0.1 below 20 THz doesn't mean that THz radiation above 20 THz is completely blocked. Apparently, the measured spectra shown in Fig. 3(b) confirm the presence of high-frequency THz radiation at 20-100 THz. Considering the strong attenuation factor (10^{-4}) of the combined filters, this scheme dominantly produces mid/long-wavelength IR prior to its detection! The measured autocorrelation traces also confirm strong mid/long-wavelength IR generation. The FWHM for acetone and ethanol is ~19 fs, and this corresponds to a radiation bandwidth (or peak radiation frequency) of ~50 THz.

Response: What we are concentrating on is the generation of THz radiation due to the in-phase mixing of the fundamental and the second harmonic (of the supercontinuum) in the liquid, and have shown that this

generation mechanism is more efficient compared to the conventional 2-color filamentation in air. Indeed, as we mention in the Methods section, there is generation of strong MIR radiation from the laser filamentation in the liquids, and therefore we employed appropriate filters to attenuate the radiation as far as possible. Moreover, as correctly pointed out by the reviewer and mentioned in our manuscript, a FWHM of 19 fs in the autocorrelation trace corresponds to a large bandwidth indeed. We can approximately estimate the percentage of spectral power at different frequency bands from Fig. 3 (b). The bar-chart below shows the spectral power percentage in the 4 frequency bands: 0.1-20 THz, 20-50 THz, 50-100 THz, and > 100 THz. For Air, 99.5% of the spectral power lies between 0.1-50 THz. In comparison, Acetone has 72% of the spectral power in the same 0.1-50 THz range. For ethanol, the number is 64%. We have observed that, on an average 70% of the spectral power is in the 0.1-50 THz range for the liquids. We use this information during calculation of the integrated energy from the liquids.

Revision in the Manuscript:

We have included the spectral power distribution information in the Results (Page 7, Line 9-12 and Line 20-21).

2. Given that the conventional THz frequency band ranges from 0.1 to 10 THz, my suggestion is to call the observed radiation as THz/mid-IR radiation (or something equivalent), not simply THz radiation, throughout the paper.

Response: In most of the recent works on THz [Kim *et. al.*, *Nat. Photonics* **2**, 605–609 (2008); Oh *et. al.*, *New J. Phys.* **15**, 75002 (2013); Gopal *et. al.*, *Opt. Lett.* **38**, 4705–4707 (2013)], the band has been extended

up to 50 THz [some even include the MIR (~ 100 THz) in the broad definition]. Our definition of THz conforms to the contemporary nomenclature (~ 0.1 – 50 THz). We have referred to the generated radiation as broadband THz in the abstract and the introductory section of the manuscript.

Revision in the Manuscript:

For the sake of clarity, we specifically mention 0.1 – 50 THz as the bandwidth in the Abstract. We also revise “THz radiation” to “broadband THz radiation” at appropriate places in our manuscript, while avoiding unnecessary repetitions. The title of the paper now reads “**Highly efficient broadband terahertz generation from ultrashort laser filamentation in liquids**”. The revised texts are marked in deep-red throughout the manuscript.

3. One related question is how the maximal THz energy ~76 uJ was estimated? Does it represent the final energy measured by the pyroelectric detector? Or is the energy before the filter combination? If so, what factor was used to estimate the energy before the filters? Also this energy represents all radiation frequencies as shown in the spectrum figure?

Response: The numbers quoted in the energy measurement is obtained as follows:

- a) The integrated radiation is collected by the pyroelectric detector in the presence of the filter combination (2×0.5 mm HRFZ-Si + Black LR-Si + 5mm HDPE). The pyroelectric detector records a voltage signal, which is converted to energy values using instrument calibration data. Say, this value is $X \mu\text{J}$.
- b) Since on an average ~ 70% of the detected radiation lies in the 0-50 THz range, the integrated broadband THz energy is estimated as $Y = 0.7 X \mu\text{J}$.
- c) Now we apply the correction for the filter combination (from their individual THz transmission data in the THz range), which we conservatively take as ~ 0.1 (for 0.1-20 THz) over the whole 0.1-50 THz range, and obtain the THz energy as, $U = Y/0.1 \mu\text{J} = 7 X \mu\text{J}$. (This is an underestimate of the energy)

Thus U (in μJ) is the estimated integrated broadband THz energy radiated from the liquid source due to femtosecond laser filamentation.

Revision in the Manuscript:

We have clarified this in our revised manuscript in the Results (Page 4, Line 7-8), and Methods (Page 12, Line 21 – Page 13, Line 2).